# celldeath: A tool for detection of cell death in transmitted light microscopy images by deep learning-based visual recognition

**Alejandro Damián La Greca**[1][◉], **Nelba Pérez**[1][◉], **Sheila Castañeda**[1], **Paula Melania Milone**[1], **María Agustina Scarafía**[1], **Alan Miqueas Möbbs**[1], **Ariel Waisman**[1,2], **Lucía Natalia Moro**[1,2], **Gustavo Emilio Sevlever**[1], **Carlos Daniel Luzzani**[1,2], **Santiago Gabriel Miriuka**[1,2]*

**1** Laboratorio de Investigación Aplicada a Neurociencias, FLENI-CONICET, Buenos Aires, Argentina, **2** Consejo Nacional de Investigaciones Científicas y Técnicas (CONICET), Ciudad Autónoma de Buenos Aires, Buenos Aires, Argentina

◉ These authors contributed equally to this work.
* smiriuka@fleni.org.ar

**Data Availability Statement:** Image data are fully available without restrictions from https://www.kaggle.com/miriukalaboratory/cell-death-in-seven-cell-lines.

## Abstract

Cell death experiments are routinely done in many labs around the world, these experiments are the backbone of many assays for drug development. Cell death detection is usually performed in many ways, and requires time and reagents. However, cell death is preceded by slight morphological changes in cell shape and texture. In this paper, we trained a neural network to classify cells undergoing cell death. We found that the network was able to highly predict cell death after one hour of exposure to camptothecin. Moreover, this prediction largely outperforms human ability. Finally, we provide a simple python tool that can broadly be used to detect cell death.

## Introduction

In the past few years there has been an increasing interest in artificial intelligence. The combination of newer algorithms for modelling biological data and increasing computational capacities have sparked an overwhelming amount of research for academic and biomedical purposes [1]. In particular, deep learning (DL) models inspired in neural networks (NN) have proved to be powerful. These models, called convolutional neural networks (CNN), employ backpropagation algorithms to reconfigure its parameters in successive layers while attempting to represent the input data [2], allowing them to classify complex and large sets of information, including digital images. Therefore, one of the most active fields is image recognition [3, 4].

Cell death is a complex event found in normal and pathological contexts [5]. For this reason, it is widely studied in biomedical research and it is a hallmark of many experiments, particularly in the context of drug discovery [6, 7]. Many different assays have been developed in the past decades in order to analyse cell death. All of them involve the analysis of particular features of a dying cell, including DNA fragmentation, cell membrane protein flipping, protein modifications, etc [8–10]. In any case, there is need for time and money in order to perform

**Funding:** This work was supported by grants to Dr. Miriuka from the National Scientific and Technical Research Council (CONICET) PIP112-20150100723 and from the Scientific and Technical Research Fund (FONCyT) PICT2016-0544.

**Competing interests:** The authors have declared that no competing interests exist.

these assays. An interesting approach by Chen and collaborators using weakly supervised CNN models demonstrated that they could confidently detect and count dead cells in bright-field images of cell cultures [11].

Recently, we published that NN can be used to classify transmitted light microscopy (TLM) images of differentiating pluripotent stem cells at one hour and even less, with an accuracy higher than 99% [12]. Hence, we demonstrated that applying DL over TLM images can be a powerful technology for specific purposes: we can identify the early stages of complex processes like differentiation or cell death, with nearly no money spent and with high precision. Experimental confirmation of these processes otherwise would require the use of an assay often involving time and money in several orders of magnitude. We are confident that our experience and that of many others will radically change the way fields in biology are engaged [13, 14].

In the present work we aimed to develop a simple tool for easy, fast and accurate classification of cell death in culture using TLM images. We believe that this tool can be used in any scientific lab running cell death experiments, particularly in those cases when massive and repetitive experimental settings are needed such as drug screening in cancer research.

## Materials and methods

### Cell culture and cell death induction

The four cancer cell lines and the three pluripotent stem cells used in this analysis were kept in a humidified air-filtered atmosphere at 37°C and 5% $CO_2$. Osteosarcoma U2OS cells and breast cancer MCF7 cells were routinely cultured in Dulbecco's Modified Eagle Medium (ref. 12430054, DMEM; Thermo Fisher Scientific, United States) supplemented with 10% fetal bovine serum (NTC-500, FBS; Natocor, Argentina) and 1% penicillin/streptomycin (ref. 15140–122, Pen/Strep; Thermo Fisher Scientific, United States), while prostate cancer PC3 cells and breast cancer T47D cells were cultured in Roswell Park Memorial Institute medium (ref. 22400089, RPMI; Thermo Fisher Scientific, United States) supplemented with 10% FBS and Pen/Strep. Induced pluripotent stem cells (iPS1 and iPS2, both previously developed in our lab [15]) and embryonic stem cells (H9) were maintained on Geltrex$^{TM}$ (ref. A1413302; Thermo Fisher Scientific, United States)-coated dishes using Essential 8 flex defined medium (ref. A2858501, E8 flex; Thermo Fisher Scientific, United States), replacing it each day. All cells were detached with TrypLE$^{TM}$ Select 1X (ref. A1217702; Thermo Fisher Scientific, United States) every 4 or 5 days depending on density. For death induction experiments, approximately $3x10^5$ cells were seeded in the 4 central wells of 12-well dishes (ref. 3513; CORNING Inc., United States), thus reducing potential border effects. The following day cancer cells were serum-deprived for 24h and then all cell lines were treated either with camptothecin $1–10\mu M$ (ref. C9911, CPT; Sigma-Merck, Argentina) or DMSO (ref. D2660, dimethyl sulfoxide; Sigma-Merck, Argentina) for the times indicated in experiments. To prevent addition of high doses of DMSO in high-concentration CPT treatments, more concentrated stock solutions were employed. Transmitted light microscopy images were taken immediately before adding the treatments and every hour until conclusion. Summarized information and further details on cell lines can be found in S1 Table.

### DNA damage assessment

Immunostaining was performed as previously described [16] with minor modifications. Briefly, cells treated with CPT or DMSO were fixed in 4% paraformaldehyde for 30min at room temperature and washed 3 times with PBS. Then, they were permeabilized in 0.1% bovine serum albumin (BSA)/PBS and 0.1% Triton X-100 solution for 1h, followed by

blocking in 10% normal goat serum/PBS and 0.1% Tween20 solution. Incubation with primary antibodies against $\gamma$H2AX (rabbit IgG, ref. ab2893; Abcam, United States) and p53 (mouse IgG, ref. ab1101; Abcam, United States) were performed overnight at 4˚C in 1:100 dilutions in blocking solution and later secondary antibody incubation with Alexa Fluor 594 (anti-mouse, ref. R37121; Thermo Fisher Scientific, United States) and Alexa Fluor 488 (anti-rabbit, ref. A11034; Thermo Fisher Scientific, United States) was done in the dark at room temperature for 1h together with DAPI. Cells were washed and then imaged on EVOS fluorescence microscope (Thermo Fisher Scientific, United States). Nonspecific secondary antibody binding was evaluated in the absence of primary antibodies. Images from four fields of three independent replicates were processed and analysed automatically using custom macro scripts (ImageJ software) to determine mean fluorescent intensity per nucleus and statistical significance between CPT-treated and vehicle-treated cell populations was evaluated by Welch Two Sample t-test using R.

### AnnexinV assay

Translocation of phosphatidylserine (PS) residues in apoptotic cells was detected with AnnexinV-FITC (ref. 556547; BD Pharmingen, United States) and AnnexinV-PE (ref. 559763; BD Pharmingen, United States) commercial kits, following instructions from manufacturer. Untreated and treated cells (CPT or DMSO) were collected from wells with TrypLE$^{TM}$ 1X (including supernatants), incubated with reagents provided in the kit and finally ran on BD Accuri Flow Cytometer. Results from three independent replicates were analysed using FlowJo (v7.6) software and statistical significance between CPT-treated and DMSO-treated cell populations from third quadrant (Q3) was evaluated by Welch Two Sample t-test using R.

### Transmitted light imaging

Cell images were captured in EVOS microscope using a 20x objective and setting light intensity at 40%. Between 30 and 50 images were taken across each of the 4 central wells (2 with CPT and 2 with DMSO) of multiwell plates (4 independent experiments) for each of the 7 cell lines described in *Cell culture and cell death induction*, avoiding field overlapping or any places with few or no cells and stored as png files. Size of these images was originally 960x1280 pixels, though we applied a short python script (image-slicer) to slice them into four parts in order to obtain four images from each one (480,640,3). This produced a total of 58596 images considering all timepoints (0, 1, 2 and 3h).

### Deep learning analysis

For deep learning training and prediction, we used fast.ai (v1.0.60), a frontend of PyTorch (v1.4). Briefly, training was done by using several different convolutional neural networks. ResNet50 architecture [17–19], however, was chosen among different options (ResNet34, ResNet101 and DenseNet121) because it rendered excellent results and it is widely known. Specifications on the CNN may be found in S2 Table. For analyses, images from all cell lines were split in four as previously explained resulting in a total of 15224 images from 1h, 15312 from 2h and 15032 from 3h treatments. We assigned an entire independent experiment (1 of 4) as the test set and then randomly divided the other 3 into 70% for training and 30% for validation. Final number of images in each set for all conditions assayed in this work are detailed in S3 Table. Pretrained model weights were obtained from available trainings on benchmark ImageNet dataset. Class activation maps (CAM) were constructed following specifications by the fastai project using CPT-treated and DMSO-treated random PSC images [20]. A python script with details on hyperparameter values used during trainings is available in

## Results

We defined a cell death model in all cell lines used in this work -three pluripotent stem cell (PSC) lines and four cancer cell (CC) lines- by incubating them with camptothecin (CPT), a topoisomerase I inhibitor. We have previously demonstrated that this molecule induces a very rapid cell death signaling in human embryonic stem cells that derives in apoptosis [21]. In each of the seven cell lines we titrated drug concentration and exposure time and took TLM images hourly in both DMSO (vehicle) and CPT-treated cells.

To confirm that these cell lines were undergoing apoptosis we performed different assays. Inhibition of topoisomerase I results in replication-dependent DNA double strand breaks (DBSs) [22], which lead to the phosphorylation of H2AX ($\gamma$H2AX) and activation of tumour suppressor protein p53 [23, 24]. Consistently, iPS1 pluripotent stem cells treated with CPT $1\mu M$ for 1.5h showed an increment in nuclear signal of $\gamma$H2AX as well as accumulation of p53 (Fig 1A). Compared to vehicle, the distributions of nuclear signals were significantly different for both marks (Fig 1B). We observed similar results in H9 embryonic stem cells and in iPS2 induced pluripotent stem cells.

Significant CPT-dependent activation and nuclear localization of $\gamma$H2AX and p53 (vs. DMSO) were also found in MCF7 cancer cell line at 6h of treatment (Fig 1C and 1D). All CC lines showed similar results between 3 and 6h of treatment with CPT. Interestingly, although CC lines generally evince high proliferation rates, they were practically unaffected by $1\mu M$ treatment with CPT and a concentration of $10\mu M$ was necessary to induce the apoptogenic signaling.

Longer treatments with CPT resulted in a steady $\gamma$H2AX and p53 nuclear signal in iPS1 and MCF7 cells compared to vehicle (S1A and S1B Fig), indicating that CPT treatment effectively triggers a sustained response to damaged DNA in both PSC and CC lines.

Apoptosis is a complex process and one of its earliest characteristic features is phosphatidyl-serine (PS) exposure on the outer side of the cell membrane [25]. Identification of PS residues on the surface of intact cells through its interaction with Annexin V protein enables detection of early stages of apoptosis by flow cytometry analysis. Treatment with CPT between 3 and 6h significantly increased the percentage of $PS^+$/7-AAD$^-$ cells (Q3) compared to vehicle in both iPS1 and MCF7 cells (Fig 1E and 1F, respectively). Positive values for each quadrant were determined using single stained and double stained untreated samples (S1C and S1D Fig).

Taken together, these results indicate that CPT treatment induced damage to DNA which eventually resulted in cell death by apoptosis in PSC and CC lines.

### CNN training and overall performance

Transmitted light microscopy images from all cell lines were taken at 1, 2 and 3h post induction of cell death with CPT. Minor morphological changes, if any, are observed by the first hour for all cell lines (Fig 2). In fact, deep and thorough observation is needed to capture subtle alterations in a few cell lines. For example, some degree of cell-to-cell detachment was registered in PSC lines as well as in T47D cells, and in PC3 cells, increased cell volume was observed in a portion of the images. However, none of these were markedly noticeable features and they were only present in a fraction of the images. Although later timepoints evinced more pronounced morphological changes (cell shrinkage, further detachment, nuclear condensation), they were not easily or readily detected without proper preparation.

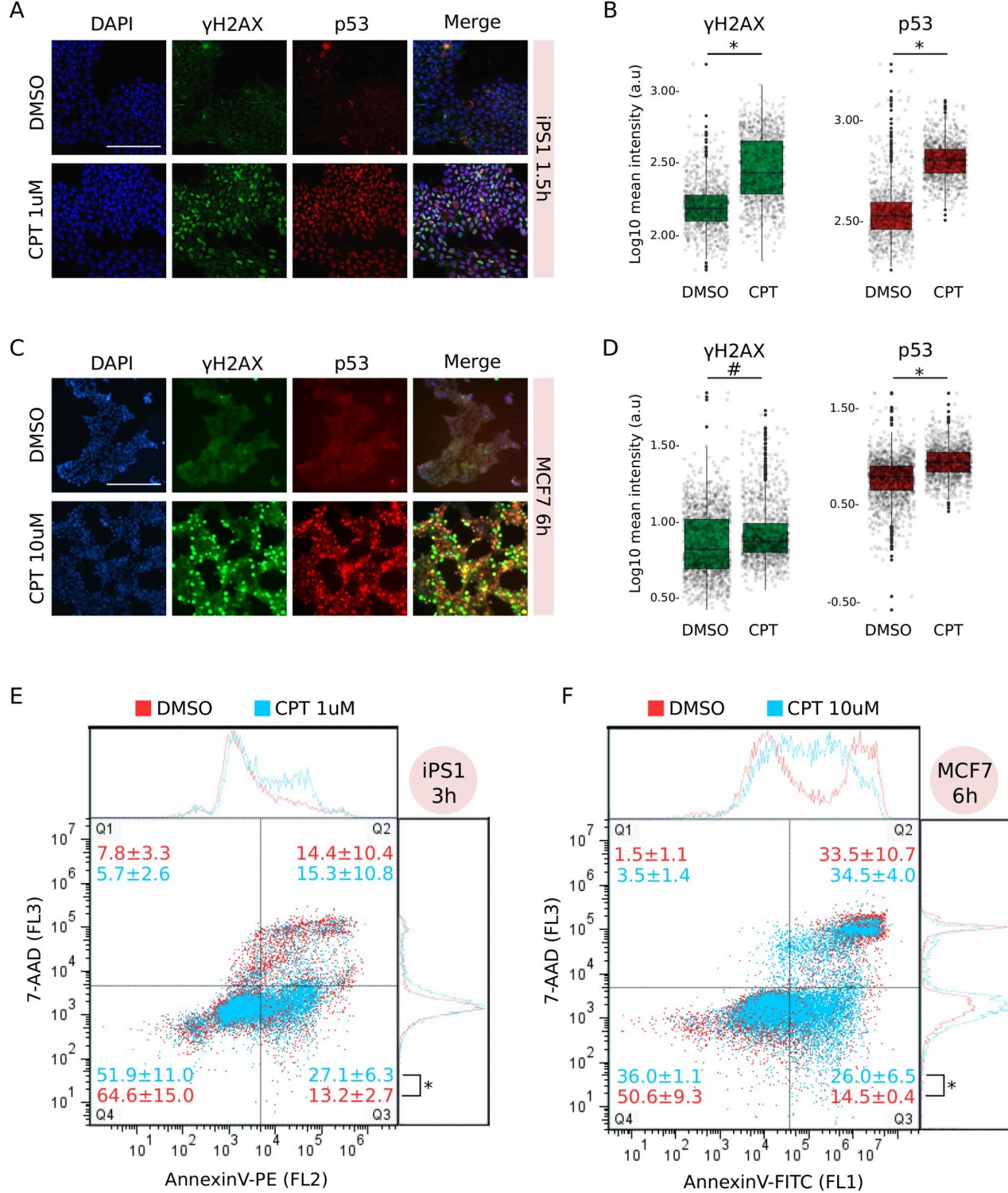

**Fig 1. Camptothecin treatment induced apoptosis in both iPS1 pluripotent stem cell and MCF7 cancer cell lines.** A) Immunostaining with anti-
γH2AX and anti-p53 of iPS1 pluripotent cell line treated (CPT 1μM) or not (DMSO: vehicle) with CPT for 1.5h. Both marks were merged with DAPI
to reveal cell nuclei and scale was set to 200μm (white bar). Images are representative of four different microscopic fields. B) Distribution of mean
signal intensity per nucleus in all fields from A, measured in arbitrary units (log10 a.u.) for γH2AX (left) and p53 (right) marks. Statistical significance
between CPT and DMSO was evaluated by Welch Two-Sample t-test (*p-value = 2.2e$^{-16}$). C) Immunostaining as in A for MCF7 cancer cell line
treated (CPT 10μM) or not with CPT for 6h. D) Mean signal intensity quantification and statistical significance were determined as in B (#p-

value = 4.89e$^{-7}$; *p-value = 2.22e$^{-16}$). E) Flow cytometry analysis with AnnexinV-PE of iPS1 cells treated with CPT 1$\mu$M (light blue) for 3h compared to DMSO (red). Incubation with 7-AAD was performed to discriminate dead cells (Q2) from early apoptotic (Q3). Number of events (cells) in each quadrant is presented as mean percentage of total population ± SEM of three independent replicates. Statistical significance between conditions in Q3 was evaluated with Welch Two-Sample t-test (*p-value = 2.5e$^{-2}$). F) MCF7 cancer cells treated with CPT 10$\mu$M (light blue) for 6h were analysed as in E, though using AnnexinV-FITC instead of PE.

Considering these minor morphological changes, we challenged 5 experienced researchers (who had never seen the images before) to correctly classify a randomly-picked set of 50 1h images (pre-training) as CPT or DMSO (vehicle). After the initial trial (without revealing performance), we "trained" the researchers by showing them 500 labelled images (CPT or DMSO) and then asked them to classify a new set of 50 images (post-training). Selection of images for trials and trainings was performed regardless of cell line or treatment. Classification performance by investigators before and after training was completely random (close to 50% correct answers), indicating that they failed to retrieve specific features which unequivocally identified each label (Fig 3A, grey bars). Moreover, decision making was mostly independent of image-related biases as very few "all incorrect" answers were registered for any given image (S2 Fig).

To assess whether deep learning-based models could outdo human performance in the early assay-free detection of cell death features, we trained a Convolutional Neural Network (CNN) using 1h CPT- and DMSO-treated images from all cell lines. The trained CNN was able to correctly classify between 9 and 10 out of 10 images in the validation and test sets (98.18$pm$0.33% and 96.56$pm$0.24% accuracy, respectively; see Methods for definition on validation and test sets) (Fig 3A, blue bars). Results presented here are based on ResNet50 NN architecture, though other architectures showed similar results (ResNet34: 98% accuracy during validation and 95% in test) (S3A Fig). While CNN robustness has been extensively tested in many situations [26], learning issues due to model set up -namely underfitting and overfitting [27]- are not uncommon and they are often associated to an unsuitable number of user-defined parameters for representing input data (too few or too many). Incremental learning of our CNN through each epoch (iterative process by which all samples in dataset took part in updating weights and parameters of the model) was diagnosed by simultaneously assessing the Loss function in the training and validation sets (Fig 3B). A minimum value in Loss function was achieved within 50 epochs, when both the training and validation sets converged at a loss value close to zero (stabilization). Extended training periods (over 200 epochs) did not dramatically improve accuracy values (S3A Fig) or loss function outcome (S3B Fig).

Learning curves (loss function) clearly showed that our model was not only suitable, but also capable of learning from input data (i.e. non-flat training curves) which is not the case in underfitted models. However, reduced generalization capabilities of the model (overfitting) are sometimes more difficult to detect considering that in fact the model is learning too well from training set. To test for this possibility we trained our model for over 100 epochs and found that the validation curve starts to increase over training curve around 280 (S3B Fig), which suggests that our model was well-fitted and only exhibited overfitting if trained for excessive periods of time.

## CNN identifies very early features of cell death

Grouping all cell lines and training the NN with only two classes (or labels), reduced potential outcomes to a binary choice between CPT or DMSO (vehicle). The final goal in this scenario was to train a model where, irrespective of cell basal morphology, the CNN was able to identify cell death. As pointed out before (CNN vs. human), successful classification at 1h was very

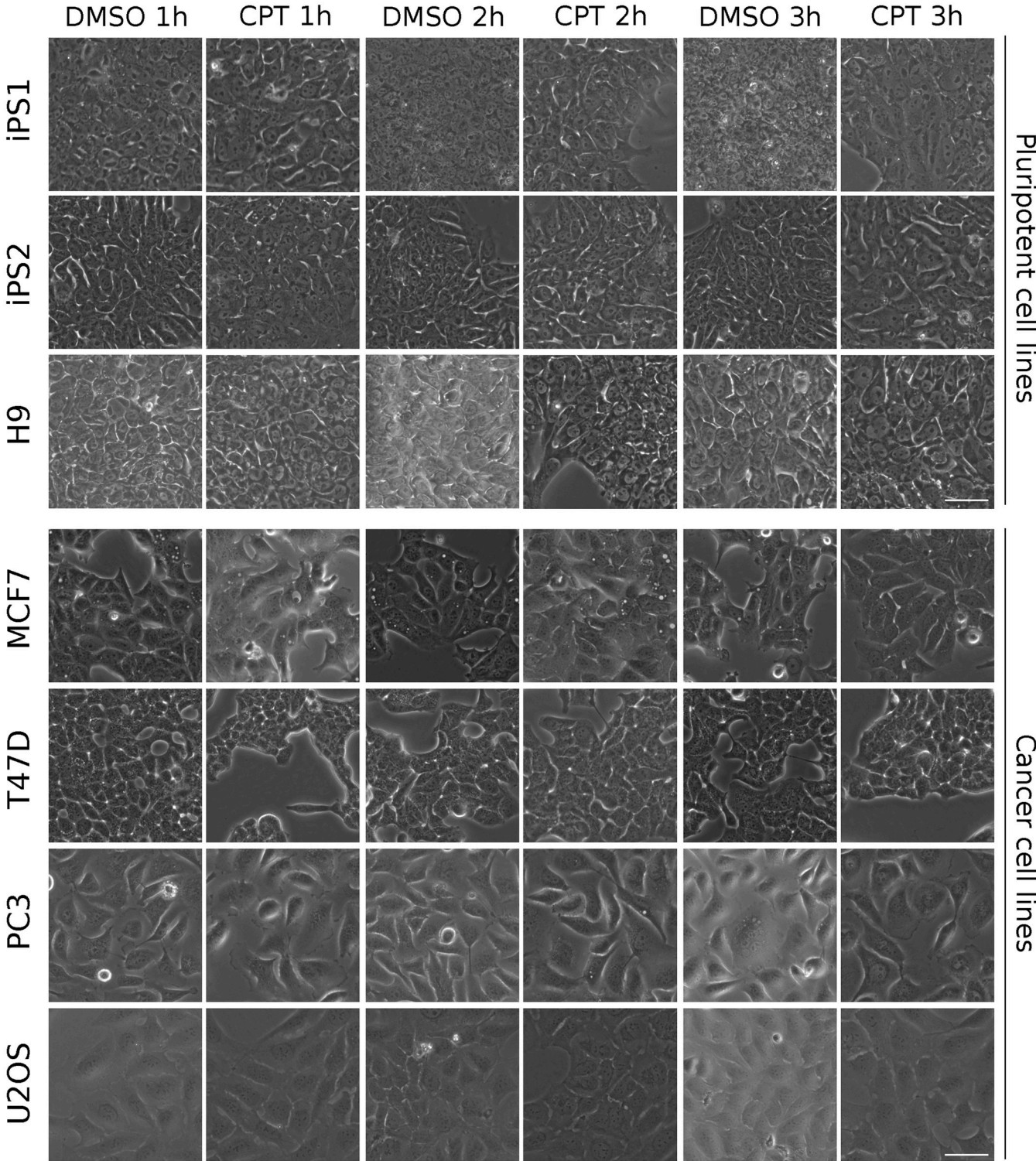

**Fig 2. Transmitted light images used for visual deep learning analysis.** Representative images of DMSO (vehicle)- and CPT-treated cell lines for 1, 2 and 3h. Scale bar is displayed in the pictures and equals to 50$\mu$m.

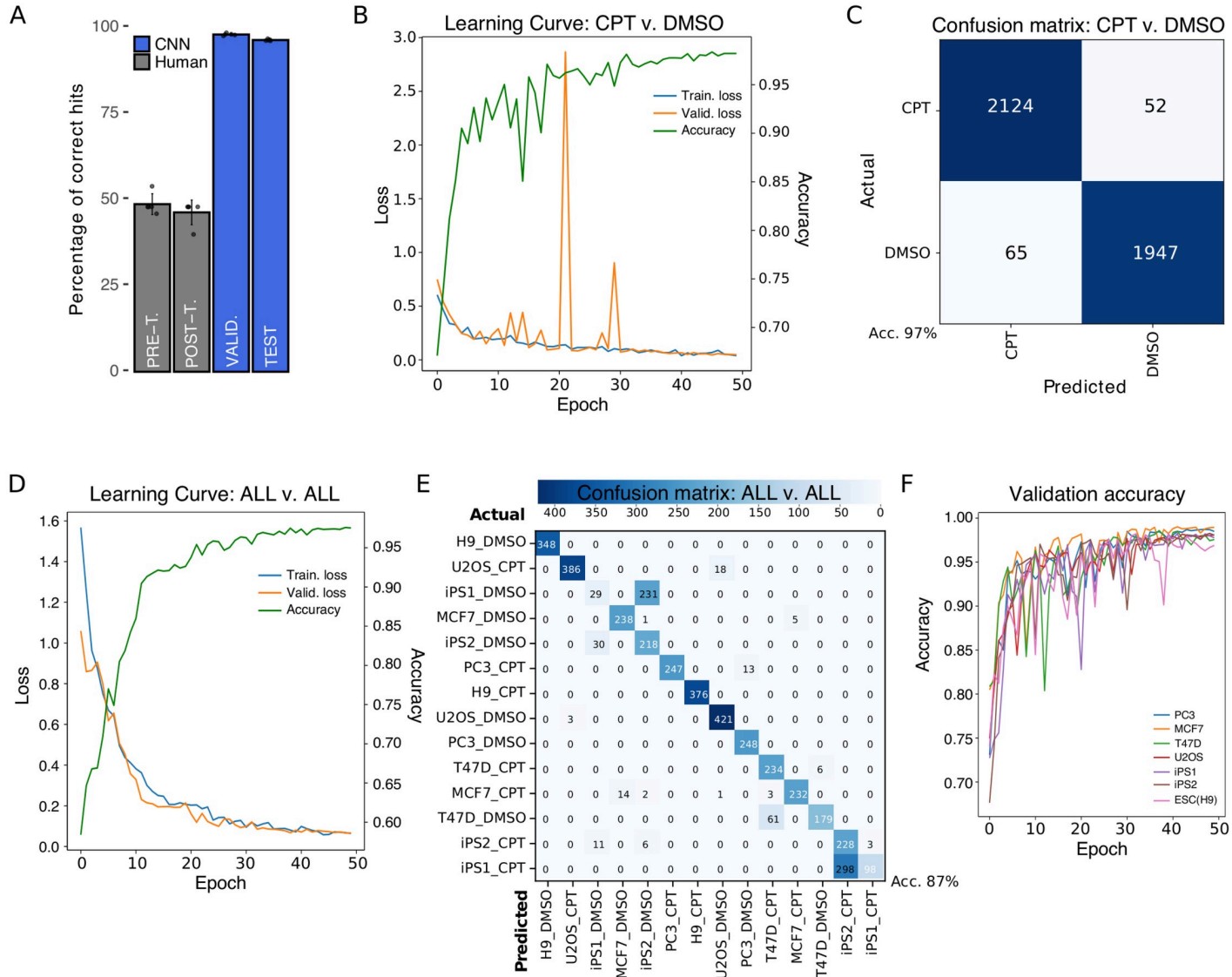

**Fig 3. Results of CNN training.** A) Comparison of human performance versus CNN. Pre (48.80±3.03%) and post-training (46.40±3.57%) results of five human subjects are shown compared to five separate runs of CNN training for a validation (98.18±0.33%) and a test set (96.58±0.24%). B) Representative Learning Curve of five independent CNN trainings using CPT and DMSO labels for 50 epochs. Accuracy curve for the same representative run is shown. C) Confusion matrix of CPT versus DMSO for training with highest test accuracy results. The highly accurate model led to very low false positives (65) and false negatives (52) during prediction on test set. D) Representative Learning Curve and accuracy of three independent CNN trainings using all cell lines and treatments as labels for 50 epochs. E) Confusion matrix of training with highest test accuracy results for all-versus-all analysis of test set. F) Validation accuracy results for training sets missing one cell line. The missing cell line was used as test set; testing accuracy for every run is shown in Table 2.

high (average accuracy of five runs in the validation set of 98.18±0.33% and 96.58±0.24% in the test set), reaching maximum accuracy values for validation and test sets of 98.67% and 97.23%, respectively, when we compared all non-exposed (DMSO) images versus all exposed ones (CPT) (Table 1). Moreover, employing a pretrained model, in which starting weights are defined beforehand rather than randomly initialized, on the same setting (imagenet CsvD) did not improve accuracy. Appropriate visual description for classification performance of our model was rendered as a confusion matrix, in which predictions on each image were contrasted to actual labels (true value). In coherence with accuracy values, the matrix showed very

**Table 1. Model performance for different conditions.**

| Condition | Train. Loss | Val. Loss | Val. Acc. | Test Acc. |
|---|---|---|---|---|
| CPTvs.DMSO | 0.068 | 0.045 | 0.9837 | 0.9723 |
| imagenet(CvsD) | 0.055 | 0.051 | 0.9825 | 0.9790 |
| ALLvs.ALL | 0.068 | 0.330 | 0.9979 | 0.8271 |
| imagenet(AvsA) | 0.029 | 0.035 | 0.9900 | 0.8658 |
| PC3 | 0.138 | 0.041 | 0.986 | 0.955 |
| MCF7 | 0.081 | 0.146 | 0.9528 | 0.9234 |
| T47D | 0.204 | 0.054 | 0.9746 | 0.8667 |
| U2O2 | 0.141 | 0.002 | 1.000 | 0.9444 |
| iPS1 | 0.379 | 0.056 | 0.998 | 0.970 |
| iPS2 | 0.091 | 0.0007 | 1.000 | 0.948 |
| ESC(H9) | 0.007 | 0.002 | 1.000 | 0.996 |

Highest value of accuracy achieved in the test set (Test Acc.) among several trainings is presented for each condition at 1h. Corresponding values of the Loss function for training (Train. Loss) and validation (Val. Loss) are shown as well as accuracy on validation set (Val. Acc.). Results of running a pretrained model on CPT vs. DMSO (imagenet CvsD) and ALL vs. ALL (imagenet AvsA) conditions were included.

few misclassification events for the total 4,188 images consisting of 65 false positives (predicted CPT, but actually DMSO) and 52 false negatives (predicted DMSO, but actually CPT) (Fig 3C). Furthermore, we found that employing the same model on longer exposure times to CPT (2 and 3h) slightly favoured an increase in validation accuracy and attenuated false detection, probably because drug-associated effects became more pronounced (S3C and S3D Fig).

To further test our model, we trained the NN to classify each cell line in each treatment (ALL vs. ALL) demonstrating a good performance as well (Fig 3D). In this case, classification was considerably improved by using a pretrained model (imagenet AvsA), with a final highest accuracy of 87% in the test set (Table 1). Although the matrix showed very few misclassification events in general, the model frequently confused DMSO-treated iPS1 for DMSO-treated iPS2 and CPT-treated iPS1 for CPT-treated iPS2 (Fig 3E), probably due to their induced-pluripotent nature. Importantly, it rarely failed to discriminate CPT from DMSO. This diagonally-populated matrix indicates that the CNN was capable of identifying cell-specific death features to correctly discriminate all labels (predicted = actual). We corroborated this finding by training, validating and testing the CNN with each cell line individually (Table 1), and again classification performance was excellent, indicating that the model can be confidently and easily applied to single or multicellular experimental layouts.

Surprisingly, we discovered that if we purposely set aside all images of one cell line during training, in some cases our model could discriminate CPT from DMSO images of that cell line during testing (Valid. accuracy ≅ Test accuracy). Even though validation accuracies were remarkably high for all training sets (Fig 3F), the model failed to accurately discriminate labels during testing with PC3 (53%) and U2OS (64%) cancer cell lines (Table 2). However, testing on the other cell lines resulted in accuracy values over 75%, particularly in PSC lines, which means that the CNN was partially able to classify images from "unknown" cells. Thus we believe that some features found useful for classification during validation might be extrapolated to unseen cell lines, but that highly cell-specific facets interfere with pattern matching. Therefore, it is always preferable that training of our model includes the cell line on which cell death prediction is intended.

**Table 2. Model performance after removing a cell line from training.**

| Cell line out | Train. Loss | Val. Loss | Val. Acc. | Test Acc. |
|---|---|---|---|---|
| *PC3* | 0.053 | 0.032 | 0.9872 | 0.5283 |
| *MCF7* | 0.054 | 0.038 | 0.9901 | 0.8688 |
| *T47D* | 0.071 | 0.047 | 0.9858 | 0.7734 |
| *U2OS* | 0.043 | 0.059 | 0.9800 | 0.6363 |
| *iPS*1 | 0.063 | 0.052 | 0.9820 | 0.9871 |
| *iPS*2 | 0.046 | 0.056 | 0.9826 | 0.9708 |
| *ESC*(*H*9) | 0.076 | 0.058 | 0.9822 | 0.9752 |

Removed cell line (Cell line out) was used for testing the model. Highest value of accuracy achieved during testing (Test Acc.) for each cell line is shown. Corresponding values of the Loss function for training (Train. Loss) and validation (Val. Loss) are shown as well as accuracy on validation set (Val. Acc.).

Finally, we analysed the images in search of features which potentially contributed the most to classification. To do so we employed class activation maps (CAM) that reconstruct heatmap-like visualizations merging the information provided by the last convolutional layer and the model predictions [20]. In other words, these heatmaps represent the score of each feature used during the decision making process as a colour-guided graphic which may facilitate human interpretation. Even though it was not clear which characteristics were in fact supporting the decision, our results demonstrate that classification was based upon features present in cell-occupied regions of the images (high activation areas) (Fig 4).

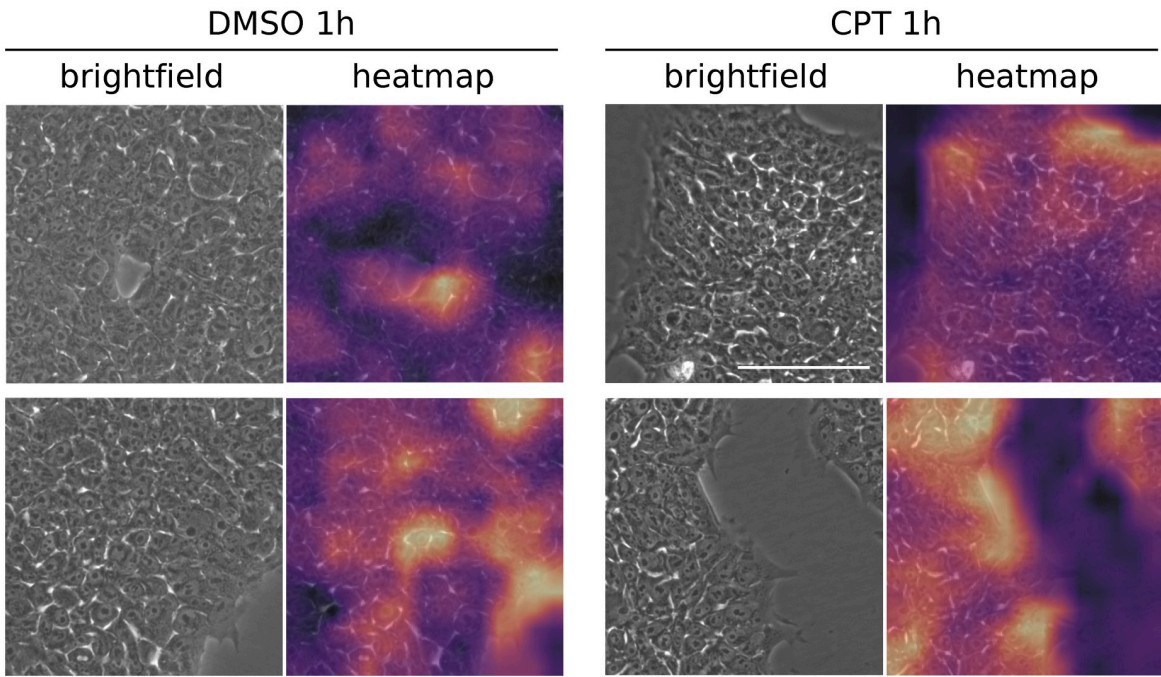

**Fig 4. Features contributing to classification.** Representative images of 1h CPT- and DMSO-treated PSC cells (brightfield) and corresponding class activation maps (heatmap). Areas in bright yellow indicate high activation for decision making and areas in purple correspond to low activation. Scale bar is displayed in the pictures and equals to 100$\mu$m.

## Discussion

Deep learning techniques are being increasingly used in the biomedical field [14, 28]. Specifically for detection of morphological changes, we [12] and others [29–32] have previously applied deep learning for different experimental approaches using TLM. For example, Ounkomol *et al* provided evidence that a DL model can predict immunofluorescence in TLM cells [30]. Jimenez-Carretero *et al* predicted fluorescent toxicity looking at changes in stained cell nuclei [32]. In a similar paper than ours, Richmond *et al* applied a CNN on TLM images in order to predict phototoxicity, but their accuracy was approximately 94.5%, probably related to the shallow network they used. Moreover, it took them 16h of training to reach this level, whereas our model gets ≅99% accuracy in approximately 3–4h using a similar hardware. Finally, they did not provide any easy way to reproduce and apply their findings.

In this work we showed that convolutional neural networks can be trained to recognize very early features of cell death. We trained the NN with images taken just after one hour of starting cell death induction, at which point the human eye was unable to identify morphological changes to correctly classify a set of images. We conducted a standard "single-blind" test in which several trained investigators from our institution assessed a set of images and attempted to classify them into treated (CPT) or vehicle (DMSO). Although we allowed them to train after the initial trial, investigators were unable to properly identify the very early changes in cell death. In fact, their results were practically random. However, their low performance may be related to the fact that any regular cell culture exhibits some degree of cell death, and actually our experiments showed that a few cells in the control group displayed translocation of annexin V (Fig 1E and 1F). While this might constitute a potential confounding factor for the researcher, it does not apparently impact on CNN learning. In the last few years there have been significant advances in whole-image-recognition approaches, but still it is not always possible to clearly identify which image features shift the balance towards an accurate classification. Although computer-vision (field of image recognition) scientists are developing new and more complex algorithms (e.g. class activation maps, occlusion, deconvolution) in an attempt to better interpret these features and correct for model biases, there is no rigorous consensus in the field on what the network might be considering relevant and how to expose it [33]. We believe that our model could be recognizing subtle alterations in cell membrane, cytoplasmic vesicles and/or changes in the nuclear morphology proper of the ongoing cell death process rather than cues from the background.

In our experiments we found that DL algorithms can reach high accuracy values for detection of morphological changes in TLM images. Particularly, PSC lines produced better test results than CC lines in all conditions, indicating that CPT-induced features are perhaps more easily recognizable in the former. Consistently, the effects of CPT treatment collected by flow cytometry and immunofluorescence were already visible by 1h in PSC lines, while it took no less than 3h and higher CPT concentrations to achieve similar results in CC lines. In line with our observations, previous results demonstrated that pluripotent cells were in fact more sensitive to CPT treatment compared to differentiated cells [34, 35] and it is also possible that the accumulation of mutations associated with cancer cell lines could have conferred some degree of tolerance against DNA damage.

Improving training results of a CNN is not an easy challenge. While it is true that implementing models based on widely known architectures (e.g. ResNet50) incorporates many standard settings and default hyperparameter values, fine-tuning a model is typically an empirical endeavour. One of the major determinants in achieving well-trained models relies on the number of samples employed in the run [36, 37]. This was clearly demonstrated when we further explored the capabilities of our model by introducing more labels to the same training set

(less images per label), which resulted in a weaker performance. Instead of the initial binary setting (CPT vs. DMSO), in this case labels included the name of each cell line as well (ALL vs. ALL) culminating in accuracy values on the test set that dropped nearly 15%. When increasing sample size is not feasible, there are still several options to enhance performance (e.g. data augmentation, learning rates adjustment). The use of pretrained models that carry weights information from training on benchmark datasets like ImageNet (transfer learning), might help to reduce training time and generalization errors (prediction) [38].

Generalization is a major goal in deep learning (i.e. predictive modelling) [2], though as a rule of thumb, to predict if "something is a dog", one must train the network with images of dogs. In that matter, our model was able to accurately classify images from a biological replicate not included in the training set (Fig 3C and 3E). Unexpectedly, it also showed a remarkable capacity to discriminate treated and untreated images from previously unseen cell lines. This suggests that the network was capable of extracting some features intrinsically associated to the cell death process and extrapolate them onto unknown but related images (the left out cell line in our case). However, these results should not be interpreted as the identification of hallmark morphological signatures of the apoptotic process. In spite of displaying neither under- nor overfitting during training, the model did not produce similar results on CC and PSC lines, which is plausibly related to the higher sensitivity of PSC lines to CPT (more distinguishable features at earlier timepoints, 1 to 3h). In fact, it should be a notable reminder that input-dependent factors (e.g. cell type-specific morphology, drug response time, pathway activation) will influence the predictive power of the model. Hence, we strongly recommend including all cell lines and conditions on which future predictions are intended to capture the complexity of input data and achieve test accuracy results comparable to validation. Adequately representing the complexity in input image data is the empirical result of balancing network performance and generalization (visit https://github.com/miriukaLab/celldeath for details on tunable parameters).

Besides the proof of concept regarding the ability of NN for cell death detection, we also provide a set of scripts wrapped in a python-based tool for a straightforward implementation of this technology. In everyday laboratory practice, this may be a significant advantage for designing and running experiments as it is possible to scale-up throughput and more importantly readout. In particular, the use of these technologies together with automation in highly repetitive assays should increase reproducibility and reduce costs. With minimal knowledge on deep learning and command line usage, any researcher can run our scripts to get results similar to ours on their own sets of images.

In conclusion, we found that DL can be applied for cell death recognition in transmitted light microscopy images and we provide a user-friendly tool to be implemented in any lab working on cell death.

## Supporting information

**S1 Table. Description of cell lines used in this work.**
(XLSX)

**S2 Table. Deep learning model specifications.**
(ODT)

**S3 Table. Number of images per condition.**
(XLSX)

**S1 Fig. Effect of longer CPT exposure times on $\gamma$H2AX and p53 staining and flow cytometry controls.** A) iPS1 cells were treated or not (DMSO) with CPT 1uM for 3 and 5h. Cells were

stained with anti-$\gamma$H2AX or anti-p53 and nuclei were revealed with DAPI. Scale was set to 200um (white bar). B) MCF7 cells were treated or not (DMSO) with CPT 10uM for 8h. Cells were stained as in A. C) Controls used for setting background levels in iPS1 flow cytometry experiments. D) Controls used for setting background levels in MCF7 flow cytometry experiments.
(TIFF)

**S2 Fig. Human trials.** Detailed results of five subjects involved in scientific activities tested for their capacity to discriminate cells treated with CPT from DMSO before (Pre-) and after (Post-) being trained with a different set of images.
(TIFF)

**S3 Fig. Neural network performance.** A) Comparison of accuracy results between ResNet50 and ResNet34 architectures using the same input data and parameters. B) Learning curve (training and validation sets) for ResNet50 architecture during extended training (400 epochs). Point of inflection in validation curve is indicated with an arrow inside the inset box. Validation accuracy for the training run is also shown. C) Confusion matrix for images of 2h CPT/DMSO-treated cells. D) Confusion matrix for images of 3h CPT/DMSO-treated cells.
(TIFF)

# Acknowledgments

We would like to thank Dr. Elba Vazquez, Dr. Adalí Pecci, Dr. Luciano Vellón, Dr. Martín Stortz and Dr. Alejandra Guberman for providing many of the cell lines used.

# Author Contributions

**Conceptualization:** Alejandro Damián La Greca, Ariel Waisman, Lucía Natalia Moro, Carlos Daniel Luzzani, Santiago Gabriel Miriuka.

**Data curation:** Alejandro Damián La Greca, Nelba Pérez, Sheila Castañeda, María Agustina Scarafía, Alan Miqueas Möbbs.

**Formal analysis:** Alejandro Damián La Greca, Nelba Pérez, Sheila Castañeda, Paula Melania Milone.

**Funding acquisition:** Santiago Gabriel Miriuka.

**Investigation:** Alejandro Damián La Greca, Sheila Castañeda, María Agustina Scarafía, Alan Miqueas Möbbs.

**Methodology:** Alejandro Damián La Greca, Nelba Pérez, Sheila Castañeda, Paula Melania Milone, María Agustina Scarafía, Alan Miqueas Möbbs, Santiago Gabriel Miriuka.

**Project administration:** Santiago Gabriel Miriuka.

**Software:** Alejandro Damián La Greca, Nelba Pérez, Santiago Gabriel Miriuka.

**Supervision:** Gustavo Emilio Sevlever, Santiago Gabriel Miriuka.

**Visualization:** Alejandro Damián La Greca.

**Writing – original draft:** Alejandro Damián La Greca, Santiago Gabriel Miriuka.

**Writing – review & editing:** Alejandro Damián La Greca, Ariel Waisman, Lucía Natalia Moro, Gustavo Emilio Sevlever, Carlos Daniel Luzzani, Santiago Gabriel Miriuka.

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
