## [Decision Letter · Decision Letter 0]

6 Apr 2021

PONE-D-21-04850

celldeath: a tool for detection of cell death in transmitted light microscopy images by deep learning-based visual recognition

PLOS ONE

Dear Dr. Miriuka,

Thank you for submitting your manuscript to PLOS ONE. After careful consideration, we feel that it has merit but does not fully meet PLOS ONE’s publication criteria as it currently stands. Therefore, we invite you to submit a revised version of the manuscript that addresses the points raised during the review process.

We look forward to receiving your revised manuscript.

Kind regards,

Chi-Hua Chen, Ph.D.

Academic Editor

PLOS ONE

Journal Requirements:

4. In your Data Availability statement, you have not specified where the minimal data set underlying the results described in your manuscript can be found (only "image data"). PLOS defines a study's minimal data set as the underlying data used to reach the conclusions drawn in the manuscript and any additional data required to replicate the reported study findings in their entirety. All PLOS journals require that the minimal data set be made fully available. For more information about our data policy, please see http://journals.plos.org/plosone/s/data-availability.

5. Please amend the manuscript submission data (via Edit Submission) to include author Paula Melania Milone.

[We would like to thank Fleni-CONICET Institute and Pérez Companc Foundation for their continuous support.]

 [The author(s) received no specific funding for this work.]

Reviewers' comments:

Reviewer's Responses to Questions

**Comments to the Author**

1. Is the manuscript technically sound, and do the data support the conclusions?

Reviewer #1: Yes

2. Has the statistical analysis been performed appropriately and rigorously? 

Reviewer #1: Yes

3. Have the authors made all data underlying the findings in their manuscript fully available?

Reviewer #1: Yes

4. Is the manuscript presented in an intelligible fashion and written in standard English?

Reviewer #1: Yes

5. Review Comments to the Author

Reviewer #1: In this work Alejandro La Greca et al., show that a neural network is able to predict cell death accurately after one hour of exposure to camptothecin. The network’s prediction meaningfully outperforms human capabilities.

The results are very significant given (1) the cumbersome procedures that must be taken to achieve the same task via fluoresce (the accepted standard), and (2) the ubiquitous application of assessing cell death in biomedicine.

However, there are two important issues that must be addressed before publication:

1. The authors should provide a more direct visual comparison between accepted measures of cell death (i.e., fluorescence based) and their network output. Fig 4 provides a very crude visualization of this, but it is impossible to tell how real the output is without a direct comparison to a ground truth.

2. The authors should provide a more meaningful discussion regarding how these results may (or may not) be extrapolated. Will a network need to be retrained for every cell type under every different condition? Or will this approach enable a more universal assessment of cell death after training with representative cells in perhaps non-identical circumstances?

6. PLOS authors have the option to publish the peer review history of their article (what does this mean?). If published, this will include your full peer review and any attached files.

Reviewer #1: No

---

## [Author Response · Author response to Decision Letter 0]

24 May 2021

We have uploaded to the editorial manager a "Response to Reviewer" file. Below you will find a copy of that letter.

Hereby we submit a revised version of our manuscript entitled “celldeath: a tool for detection of cell death in transmitted

light microscopy images by deep learning-based visual recognition” by Alejandro La Greca and collaborators. We have

addressed all concerns and requirements indicated by the Journal, as well as comments raised by Reviewer 1 in a

point-by-point fashion.

Modifications introduced to the manuscript are properly declared in the“Response to Reviewers” letter and they are

clearly marked in the “Revised Manuscript with Track Changes” pdf file. Also, a revised unmarked “Manuscript” pdf file

has been included with this submission along with the LaTex source file.

Journal Requirements

1. Please ensure that your manuscript meets PLOS ONE’s style requirements, including those for file naming.

As a result of using PLOS ONE’s latex template, our manuscript follows all necessary style requirements.

2. Please review your reference list to ensure that it is complete and correct.

Reference list was properly checked.

3. PLOS requires an ORCID iD for the corresponding author in Editorial Manager on papers submitted after

December 6th, 2016.

We have associated the ORCID iD of Dr. Santiago Miriuka (corresponding author) to this submission.

4. In your Data Availability statement, you have not specified where the minimal data set underlying the

results described in your manuscript can be found (only ”image data”).

The minimal data set to reproduce all CNN trainings and tests can be found in https://www.kaggle.com/miriukalaboratory/celldeath-

in-seven-cell-lines.

5. Please amend the manuscript submission data (via Edit Submission) to include author Paula Melania

Milone.

Manuscript submission data has been amended to include author Paula Melania Milone.

6. Please remove any funding-related text from the manuscript and let us know how you would like to update

your Funding Statement.

Funding-related text has been removed from manuscript. Funding Statement should go as follows:

This work was supported by grants to Dr. Miriuka from the National Scientific and Technical Research Council

(CONICET) PIP112-20150100723 and from the Scientific and Technical Research Fund (FONCyT) PICT2016-

0544.

Reviewers’ comments

Reviewer #1

We are grateful for Reviewer #1’s comments and insightful remarks. Issues raised by Reviewer have been addressed

below.

1) The authors should provide a more direct visual comparison between accepted measures of cell death (i.e., fluorescence based)

and their network output. Fig 4 provides a very crude visualization of this, but it is impossible to tell how real the output is

without a direct comparison to a ground truth.

We believe that Reviewer #1 raised a reasonable point considering that we proposed that our model markedly outperforms

humans and could eventually replace laborious, expensive or repetitive experimental procedures. While it is true

that Fig4 could further explore the features regarded as relevant to the network, we did not attempt to construct an

explanatory model from our trained CNN (see Shmueli et al. 2010 for further details on this matter). In fact, it would

be inaccurate to decode features responsible for classification through predictive models -such as neural networks-. As

a consequence, the signal observed in the class activation maps will not necessarily correlate with any fluorescent mark.

At present, however, state-of-the-art computer-vision (field of image recognition) algorithms are being developed in an

attempt to better interpret “what is the network actually seeing” and perhaps reduce model biases and generalization

errors (Guidotti et al. 2019). Our intention with Fig4 was simply to demonstrate that our model was extracting features

potentially important for classification from cell-occupied regions of the images rather than background.

We have included an explicit reference to this issue in revised manuscript (Discussion, paragraph 2).

2) The authors should provide a more meaningful discussion regarding how these results may (or may not) be extrapolated.

Will a network need to be retrained for every cell type under every different condition? Or will this approach enable a more

universal assessment of cell death after training with representative cells in perhaps non-identical circumstances?

In agreement with Reviewer’s suggestion, we have included a more significant discussion on this topic in the revised

manuscript (Discussion, paragraph 5). Also, we revised the corresponding text in Results section to provide clearer

statements (Results, subsection “CNN identifies very early features of cell death”, paragraph 3).

Regards,

Dr. Santiago Miriuka

---

## [Decision Letter · Decision Letter 1]

10 Jun 2021

celldeath: a tool for detection of cell death in transmitted light microscopy images by deep learning-based visual recognition

PONE-D-21-04850R1

Dear Dr. Miriuka,

We’re pleased to inform you that your manuscript has been judged scientifically suitable for publication and will be formally accepted for publication once it meets all outstanding technical requirements.

Kind regards,

Chi-Hua Chen, Ph.D.

Academic Editor

PLOS ONE

Additional Editor Comments (optional):

Reviewers' comments:

Reviewer's Responses to Questions

**Comments to the Author**

1. If the authors have adequately addressed your comments raised in a previous round of review and you feel that this manuscript is now acceptable for publication, you may indicate that here to bypass the “Comments to the Author” section, enter your conflict of interest statement in the “Confidential to Editor” section, and submit your "Accept" recommendation.

Reviewer #1: All comments have been addressed

2. Is the manuscript technically sound, and do the data support the conclusions?

Reviewer #1: Yes

3. Has the statistical analysis been performed appropriately and rigorously? 

Reviewer #1: Yes

4. Have the authors made all data underlying the findings in their manuscript fully available?

Reviewer #1: Yes

5. Is the manuscript presented in an intelligible fashion and written in standard English?

Reviewer #1: Yes

6. Review Comments to the Author

Reviewer #1: (No Response)

7. PLOS authors have the option to publish the peer review history of their article (what does this mean?). If published, this will include your full peer review and any attached files.

Reviewer #1: No

---

## [Editor Report · Acceptance letter]

15 Jun 2021

PONE-D-21-04850R1 

celldeath: a tool for detection of cell death in transmitted light microscopy images by deep learning-based visual recognition 

Dear Dr. Miriuka:

I'm pleased to inform you that your manuscript has been deemed suitable for publication in PLOS ONE. Congratulations! Your manuscript is now with our production department. 

Kind regards, 

on behalf of

Professor Chi-Hua Chen 

Academic Editor

PLOS ONE